# "Making" Waves: How Young Learners Connect to Their Natural World through Third Space

**Anne Burke** [1,*] **and Abigail Crocker** [2]

[1] Department of Education, Memorial University of Newfoundland, St. John's, NL A1B 3X8, Canada
[2] Department of Folklore, Memorial University of Newfoundland, St. John's, NL A1B 3X8, Canada; abcrocker@mun.ca
\* Correspondence: amburke@mun.ca; Tel.: +1-709-864-8610

**Abstract:** In a world that grows increasingly aware of ecological problems such as global warming, rising sea levels, and pollution, we need to reconsider how we connect ourselves to the natural world around us. In this paper, we view makerspaces as ideal locations to shape children's emotional, sociocultural, and educational consciousnesses about the environment and our multi-layered roles undertaken to live in, and conserve, it. We apply third space, makerspace, and relational value theories in the analysis of a research project conducted with children at an early childhood centre. This project invited children to discuss ocean conservation prompted by the picturebook *Flotsam* (2006) and create three-dimensional exhibits that express how they visualize ocean conservation. Our research shows that children develop strong emotional connections to tangible representations of conservation when they are given the time to invest in making them, and that these emotional connections are driving forces for relational values that create conservation-oriented mindsets. It also shows how important context is for shaping the ways children learn, and that providing opportunities to examine conservation through makerspaces as a third space encourages children to create empathetic and personal relationships with the natural world.

**Keywords:** early learning; makerspaces; maker literacies; third space theory; relational values; environmentalism; conservation

---

## 1. Introduction

Makerspaces include elements of constructionism, as defined by Papert (1980), which involves an inquiry-based approach to knowledge creation [1]. Through processes of construction and deconstruction in a purposefully designed environment, children's play becomes a vehicle for constructionist learning. Makerspaces' digital capacities enable self-directed and collaborative learning, moving from materials consumption to digital creation tools [2,3]. Complex terms or processes such as sustainability and the environment can be explored in the maker process through manipulation of play environments and loose parts that contribute to multimodal meaning making. For young children, the playmaking and creative possibilities in a makerspace enables children to engage in more imaginative possibilities exploring more challenging societal concerns through digital and maker literacies [4–6], computational skills [7,8], and STEM concepts [9–11]. The manipulation of digital tools and crafting materials in the maker process personalizes the learning process, as children control and direct the path of learning.

Makerspaces allow children to leverage their own sociocultural notions of reality from family and home environments to inform the path of their learning [12]. Such technologies, utilizing digital and multimodal elements provide a fluid multi-disciplinary learning environment almost exclusive to makerspaces, creating new possibilities for innovation and multiple entry points for participants [13].

Further consideration of how digital literacies in combination with traditional literacies can support children's meaning making is essential in a growing digital world. In this article, we show how makerspaces leverage both children's existing sociocultural and academic knowledge to inform the creation of new environmental knowledge. In our research, we consider makerspaces as a viable third space (see Oldenburg 1997) that empowers children's voices through making [14]. In this article, we present a case study conducted at an after-school early learning centre, where we maintain that makerspaces are an ideal third space to explore children's relational values in consideration of ocean conservation [15]. We frame this data through Oldenburg's 1997 conceptualization of third space for children's environmental learning, and further consider the emotional entanglements of young children's voices through the use of David Wiesner's picturebook *Flotsam* [16]. We consider how a makerspace provides the ideal backdrop in the exploration of third space through the eyes of children, early childhood educators and pre-service teachers while focused in maker literacies exploring ocean sustainability. Our project considered these two research questions: How could maker literacies, materials and play-making in a makerspace help children build a relationship with nature, and how can a makerspace as a third space aid children's emotional connections and responsibility for ocean conservation?

## 2. Structure of Makerspaces

Makerspaces are surging in popularity in schools, libraries, and other educational institutions as educators recognize a makerspace's potential for construction-based learning. Collins (2017) notes that "the philosophy behind makerspaces is that it is a place for students not just to play but play constructively in an environment that encourages curiosity, exploration, risk-taking, and creative freedom" [17] (p. 15). The role of the makerspace in education is to consciously provide children with opportunities to learn from their own experiences through their construction, deconstruction, and manipulation of both digital and papercraft resources. Through this process of creation, children develop emotional attachments and attach meaning to the subjects of their work. These makerspaces can be physical or virtual spaces where computers are used by children to expand their play to create new digital media, or learn through digital platforms that explore various concepts in STEAM (science, technology, math, art, and engineering) subjects. Such spaces can be tactile or mechanical, involving the use of fabric, paper, machines, or technology. Makerspace play can be wholly child directed, or the teacher may guide children with explanations of materials, machines, or modules, so that they are more informed about the resources of the space.

Most often, makerspaces invite play-based learning through inquiry-based activities of active and involved learners [18]. In this case study, our research explored the learning that takes place at the intersection of both digital and non-digital makerspace settings. Children, early childhood educators, and pre-service teachers worked together to talk about ocean sustainability, over a six-week period meeting 3 to 6 h per week, and were aided by materials such as the illustrations from picturebook *Flotsam* [16]. Children created their ocean floor clay biodomes first through hand drawings using their imaginations, prior knowledge, play, and were also inspired by the illustrations and story narrative in the picturebook *Flotsam* [16].

### 2.1. Maker Literacies

Our investigation of how young children build lasting connections to nature through learning is informed through a recent growing interest in studying the role of "making" and its applications and implications regarding how children learn about ocean sustainability. One 2015 NMC Horizons Report asserts that makerspaces have "the potential to empower young people to become agents of change in their community" [19] (p. 39), but Marsh (2017) notes that this is largely dependent on what types of experiences children have within makerspaces, and how their learning is expanded when they leave this particular context [4] (p. 7). Importantly, these types of contextual experiences share a learning process whereby children came to an idea through their own creative making where

there is joy in the playmaking process which is often paired with empathy around the ideas of ocean sustainability. Furthermore, when children create as a collective, they build a shared memory that Kristiina Kumpulainen (2017) argues encourages learning and communication across disciplines, ages, learning styles, skills, and genders. Furthermore, it can be said that making "accounts for a complex set of socially and materially mediated practices that encompass not only processes of creating specific artefacts supported by a wide range of technologies and media, but also emotional, relational and cultural processes surrounding their use and construction" [20] (p. 14). It is not only important to remember that a physical connection to a subject through makerspaces enhances an emotional connection to nature but also that this process is positively amplified when conducted in a group that shares a similar narrative. In this research project focused on ocean sustainability, we saw how the physical process of making sea creatures led to an emotional connection that significantly contributed to identity development and children's relationships to nature. In particular, how children consider their own ideas situated within a makerspace collective experience, and how to negotiate successes or challenges that might affirm or conflict with a child's own experiences in another third space such as in school or at home.

Because of this, it is important to note that the context for learning is also an important catalyst towards the creation of identity. Makerspaces are sites for situated learning. Barthel et al. (2018) define 'situated learning' as learning that is "inseparable from experience" [21] (p. 3) that allows children to develop multiple expressions and emotional connections to environmental causes to further develop their environmental identity. We suggest as much as Clayton (2003) advocates that having an environmental identity "is one part of the way in which people form their self-concept: sense of connection to some part of the nonhuman natural environment" [22] (p. 45). Developing this identity, one that empathetically connects children to their natural world, also assures the longevity of global environmental conservation efforts while assisting children to expand their creative skills—both tangible and intangible—that could evolve and advance the protection of our natural world as children grow. It is also closely tied to affordance theory where "children are most likely to stay attentive and engaged when perceiving affordances that provide them with immediate, pleasurable experiential feedback about the affects of their actions" [21] (p. 3). Affordance theory is in turn closely tied to ecological psychology, which is interested in "environmental learning and action in every setting" [21] (p. 3).

## 2.2. Third Spaces

Potter and McDougall (2017) also consider the way a space is constructed to be of critical importance. Meaningful objects, student and teacher bodies, even the orientation of the seats and desks are all factors that contribute to the creation of a third space, one that sits in between the screen and the traditional classroom, and even between the home and the school space [23].

Much of the current theory about third spaces was established by sociologist Ray Oldenburg in his 1997 book *The Great Good Place*. Oldenburg originated the idea that there are third spaces, spaces that open up between home and work, places where various communities can gather, ideas can be generated, all facilitating "joyful association in the public domain" [24] (p. 1). Our project took place in an after-school programme, which is guided by child-driven curiosity, and inquiries that are often explored through playful invitations [18]. In a very real way, it is a third space—one that Ray Oldenburg defines as "a generic designation for a great variety of public places that host the regular, voluntary, informal, and happily anticipated gatherings of individuals beyond the realms of home and work" [14] (p. 16). The after-school makerspace creates and highlights multiple disciplines of learning, by combining children of different skill levels, disciplines, and abilities together in a neutral space of learning equity. The curriculum is focused around child centered inquiry learning exploring themes such as the physical environment, ocean sustainability and climate control. These contribute to the creation of physical touch, empathetic connection, and identity development. Oldenburg defined third spaces as places that serve the community best when they are *inclusive* and *local* [14] (p. xvii)—both criteria that are fulfilled by the Early Learning Centre where the research took place.

Potter and McDougall (2017) also observed that third spaces in formalized educational spaces rarely happen organically—the teacher negotiates a time and space to welcome in more student-centric appeals towards learning such as inviting professional artists into the classroom, or removing children to a more student-lead learning space such as a museum. They also note that third spaces exist most often outside of school hours in after-school programmes or during lunch periods—times when children are able to socialize without direct instruction, and draw connections between what has been taught or explored in creative and innovative ways [23]. We too drew this connection in our research. The project was conducted during an after-school programme. Though the centre is open all day to children, the programming that drew our specific interest only happened in the afternoon. It is also important to note that the project we constructed was based on the centre's curriculum related questions: How do young children build a connection with nature when they spend so much time in a classroom? How can makerspaces aid children's connections to nature? What role do pre-service teachers play when engaging young children in a makerspace? This inquiry approach involved both early childhood educators and pre-service teachers as guides for children through the group settings of their project. Because of Potter and McDougall's (2017) assertions described above, we can still consider this learning to be set within a makerspace, one which is also defined as a third space.

Makerspaces can serve many imaginative and practical functions, but they are also capable of becoming that third space. Moje et al. (2004) assert that the third space "can be viewed as a space of cultural, social, and epistemological change in which competing knowledge' and discourses' of different spaces are brought into the conversation" [25] (p. 44). Our research considered how a community of children bring voice and agency within a makerspace to improve their literacy learning about ocean sustainability and conservation, specifically by combining identities, prior knowledge, ideas, and experiences. In this way, our research both considers the type of learning that occurred within the Early Learning Center following Moje et al.'s (2004) observation—we further contextualize the learning through Situated Learning as outlined by Barthel et al. (2018)—and fills the research gap identified by Potter and McDougall (2017), specifically concerning how the collaborative nature of the third space and the free-thinking area of the makerspace may encourage young children to grapple with deeply complex topics in a meaningful way. Grappling with complex issues builds a long-lasting emotional connection to the lessons children learn and further develops an identity of compassion and conservation in a collective narrative amongst peers.

### 2.3. Manners of Learning

A common theme that connects all makerspaces is the manner of learning that is enacted within them—senses of creativity, tinkering, problem-solving, critical thinking, purpose, self-fulfillment, and happiness. We have observed that children create objects through making—not because they are coerced or forced into making. In our research, we noted children reveling in the enjoyment of making within a space that is attached neither to the responsibilities of the workspace or home-space, but one whose defining characteristics are similar to Oldenburg's third spaces. A third place is used "to signify what we have called 'the core settings of informal public life'" [14] (p. 16) that serves to be "the people's own remedy for stress, loneliness, and alienation" [14] (p. 20).

For children, we note that third spaces are different. Where an adult could go to the library, restaurant, or park without supervision, children are not often allowed the same degree of autonomy. Spaces between the home and their workspace—school—are limited. In order to enjoy the comparable experiences of joviality, relaxation, and freedom that adults do, children's third spaces must allow children the creative freedom to explore themselves and the world around them—a freedom also characteristic of makerspaces, in which they "prescribe a model of learning-by-doing in which individuals can work on creative design projects that are personally and/or collectively meaningful" [4] (p. 14). An after-school programme with a makerspace, such as the one used in this project, is one of the few places where children can experience some semblance of third-place freedom.

Through the application of Oldenburg's third space theory, our research proves that makerspaces are third spaces that considerably contribute to how children develop natural identities through making and empathy. Moreover, makerspaces are ideal locations that foster student's deep thinking, leading to empathy in ways that are critical when developing a positive relationship with the natural world. The vibrant, immersive experiences children in our research undertook showcased all eight characteristics of a third space that Oldenburg outlines: Neutral, Leveling, Conversational, Accommodating, Regular Attendees, Low Profile, Playful, and Home-Away-From-Home [14]. Through the third space provided by a makerspace we observed that children constructed the strong emotional connections needed to be life-long, conservation-minded community stewards.

## 3. Our Research

In 2018–2019, we conducted a research project in partnership with an early childcare after-school programme, and early childcare educators who had been creating the centre's digital literacy curriculum focused around the use of applications (Apps) for early learning.

Our research site as a third space has characteristics that show it exists at the intersection of home and school. Large colorful tiled floors offer the freedom of movement one finds in a dance studio with the potential of a blank canvas. Children's art covers the walls, and off to the side of the rooms are burdened shelves that showcase books, puzzles, games, arts and crafts materials, musical instruments, costumes, puppets, and gadgets that children can play with. It is a comparatively unstructured space where children can create, define, discover, and categorize social interactions with others in similar ways that adults would in cafes or parks. Children can sit on the floor or in beanbags, comfy chairs, or at tables with their activities. They can play at the sand or water tables, or in and around play houses. Digital opportunities are ever-present through digital making on tablets aided by visits of local specialists in the fields of engineering and computational coding. In his book, Hatch (2014) defines a makerspace as a "center or workspace where like-minded people get together to make things" [26] (p. 13). In an educational context, a makerspace could be a classroom corner filled with paper, string, tape, glue, blocks, scissors, magazines, milk cartons and plastic containers that children could use to create. A makerspace could just as easily be an entire room full of hardware, power tools, metal sheets, foam core, stacks of lumber and old computers, phones, radios, and TV equipment ready to be unmade and remade with aid from augmented digital technology. Makerspaces can exist in formalized education spaces, but they can also exist outside of these spaces within public libraries, museums, early childhood centres—anywhere that invites a more diverse and less organized making experience that creates that space in between home and the school classroom—the third space.

For the purposes of our research, we developed this project with Early Childhood Educators (ECEs) so that children participants in the after-school programme could explore "making" with a focus on ocean conservation under the tutelage of pre-service teachers mentored by ECEs.

All children were already attending the early learning center and all researchers had previously worked with the centre on a number of projects around their digital initiatives. Makerspaces and ocean sustainability were being explored by the children. A part of the building digital literacies initiatives was promulgated with the early childhood educators' leadership through the after-school programming leading to a viable learning partnership for pre-service teachers to be mentored by ECEs using play-based learning with digital inquiry projects at the after-school programme. Pre-service teachers in primary education training programmes were invited to share picturebooks that were focused on environmental themes. They visited the children at the centre over a six-week period and worked with the children and teachers 3–6 h weekly during this period. Early childhood educators aided the training teachers through a modelling of the reading of picturebooks to the children, encouraging discussion and reflection on the part of the children. Working with the early childhood educators, the children then were encouraged to figure out ways to approach the environmental issue through play-based learning in the after-school makerspace. In this case, the focus of this paper relies

on one group of pre-service teachers, early childhood educators and their children who experience the wordless book *Flotsam* [16].

Establishing a connection between the experiential learning of children, and the context where such learning experiences occur, is a cornerstone of a third space. Chan, Gould, and Pascual (2018) argue that eudaimonic values—values that are associated with the happiness that comes from pursuing a life of excellence regardless of the outcome—are relational values. These values are an essential component that drives children's learning within the third space context of a makerspace, and leads children to develop an emotional connection towards their learning goals, invest in the journey of discovery, which ultimately deepens their learning. This is supported by Chan, Gould, and Pascual (2018), who note that "insofar as the relationship takes on its own meaning as more than a means to an end, the thing is not wholly substitutable and the value is also relational" [15] (p. 4).

## 3.1. Research Ethics

All ethics protocols were followed as per the expectations and guidelines of our university and all participants, including the children, gave their informed consent prior to participation. This included signing up their names on a participation list located in the makerspace area that focused on a particular environmental issue. Children could pick from topics such as oceans conservation, protecting our forests, animals and habitats, and climate change. The study was conducted in accordance with the Tri-Council Policy Statement of Canada around Ethical research.

Lundy and McEvoy argue that "article 12 of the CRC, uniquely within international law, gives children the right to not only express their views but to have those views taken seriously in all matters affecting them" [27] (p. 131). Key to the success of our makerspace, and the research that was conducted within it, is the agency of the children who participate. Oldenburg's Third Place theories would also dissolve if it were not for the willing participation of individuals within them regardless of their age. We concern ourselves therefore not only with the children's participation in our research, but their right to participate. Recognizing that children are capable of advocating for their own needs and opinions with their own voices, choices are paramount to understanding how makerspaces are created, their uses as spaces that can facilitate conservationist ideologies and empathies, and their careful identification as a participatory third place.

Our aims at inclusivity, and our observations about how children learned within a space that radiates equity, are bolstered through Professor Laura Lundy's model of child participation—wherein a child's right to express their own views is supported by the right to have those views upheld and encouraged. Lundy's model argues that four elements (space, voice, audience, and influence), enacted in chronological order, facilitate the education and development of decision-making members of a community.

Our use of Lundy's model illustrates its efficaciousness when, in particular, children are given "safe, inclusive opportunities to form and express their own views" [28,29] (p. 21) such as in a makerspace. A makerspace for our research project is also enacting third-place properties; children feel comfortable expressing themselves during conversations—much like Oldenburg observes in third spaces. Though our research allowed children time to converse informally amongst each other, they were also guided through a process of academic explorations with pre-service teachers who facilitated discussion about ocean conservation employing the book *Flotsam* [16] to spark conversation. During this time, children's views and prior experiences were acknowledged, understood, and listened to by not only the pre-service teacher but by peers.

Listening to, acknowledging, and bringing others to understanding helps coalesce a gathering of individuals into a community. It sets worth to personal ideas, epiphanies, and opinions that allow a child to value how and what they think. "Knowledge," outlines Lundy and McEvoy, "is socially constructed, a position we suggest is integral to a children's rights-informed approach" [27] (p. 132). For us, the children's rights-informed approach allowed children to engage in the process of research creation by connecting logic and emotions by encouraging them to act on their new ideas, such

as with outrage, disappointment, or empathy, when the pre-service teachers during a play-based learning session purposefully "polluted" a make-believe ocean. In our research, children's voices and views were considered by pre-service teachers who observed in their reflection journals that the vivaciousness, capacity, and brilliance of each child's input far exceeded any expectation they had confirming Lundy's assertion that children are not only "*able* but also *entitled* to be engaged in [the research process]" [27] (p. 129). Such startling amazement on the part of the pre-service teachers is evidence that children were allowed and encouraged to dissect, interact with, expose, and interrogate views about conservation and how to protect the ocean—that the children's views were listened to, appreciated, and validated.

Makerspaces, when viewed as third spaces, not only act as catalysts that help develop environmentally friendly ideologies; they are key settings that enable children to exercise their distinct and natural right to participate in, form, and affect change in a research environment [30,31].

Considering children as important participants within our research project much like with Lundy and McEvoy, and establishing the research within the setting of a makerspace, gave the children the confidence to create and give voice to their prior experiences. Whilst connecting them to their current learning experiences and engaging in the process of creation—on a research level, on an artistic level, empathetically, and with a consideration towards conservation.

"When children are participants in research studies, they continue to enjoy a right to have assistance in the formation of their views (CRC, Article 12), through access to information (CRC, Article 17) and guidance from adults (CRC, Article 5)" [27] (p. 136). These researchers argue and our research supports that these views are formed freely by individuals within the group setting—expressed as evidence by the fact that only one pair of students elected to showcase mirroring dioramas where one was polluted and the other was pristine. It is in concert with their peers, when they can explore a wide range of differing opinions and complex topics that Lundy and McEvoy suggest children present their own ideas confidently without being "led to a predetermined, arguably adult, conclusion" [27] (p. 140).

## 3.2. Qualitative Case Research

Qualitative research is used to understand phenomena from the perspective of a particular population group [32]. In our case study, we share the perspectives of children on ocean conservation inspired through the use of a children's environmental literature book while engaging in activities in a makerspace. In this case, the research may be considered unique and valuable because it is supported by the children's opinions and perspectives on ocean conservation, and further substantiated through the emotional connections made to the picturebook *Flotsam* [16].

Yin (2009) reports "A case study is an empirical study that investigates a contemporary phenomenon in depth and within its real-life context, especially when the boundaries between phenomenon and context are not clearly evident" [33] (p. 18.) In this case, we were interested in how makerspaces as a third space may inform future practice of ECEs and pre-service teachers as well as make a contribution to this area of research. Our research also had us working closely with ECEs in curriculum associated with digital literacies and makerspaces and speaks to how "the concrete and applied nature of case study research makes it particularly useful for educational and curriculum researchers striving to evaluate programs, analyze specific curricula, and improve teaching practices" [34] (p. 103).

We were particularly cognizant of the children, ECEs, and pre-service teachers in our accurate representation of participant voices as we read through focus group data, interviews, observations and field notes while revisiting with theory. In particular, to ensure accuracy about "the voices and placement of our respondents within the text" [35] (p. 7). When discussing research participant representation, Given (2000) submits that "more unstructured methods such as unstructured interviews and diaries let participants express their experiences in their own natural language and setting, thereby giving them full voice" [36] (p. 756). We drew from pre-service teachers and ECEs daily observation logs, and diary entries of their experiences during the project to frame our understandings of makerspaces as

third spaces that develop children's empathy and relationship with nature. As researchers, "we conduct qualitative research when we want to empower individuals to share their stories, hear their voices, and minimize the power relationships that often exist between a researcher and the participants in a study" [37] (p. 48). In particular, we wished to create a rich dialogue in this case study, which provides a fine grain account representing the contexts of children's understanding of ocean conservation when experienced in a third space makerspace.

## 4. Analysis of the Project

Within the context of our project, both pre-service teachers and the children discovered that pursuit of learning is of equal or of greater importance than the outcome, and therefore the experiences all learners gained and the relationships shared with the concepts of conservation, and ocean sustainability are therefore more valuable. We found that children developed shared relationships not only with their peers, but with their teachers, their learning experiences, and the subject material—all values that Chan, Gould, Pascual (2018) argue are relational, and that by their nature, function best within the context of third spaces [15]. Brown (1984) states that "value in this realm is not observable" [38] (p. 233) but we observed through our project that relational values are best enacted within a third space and can contribute to an emotional connection that deepens children's learning and contributes to positive relational values that can "because of their somewhat unique combination of groundedness and moral relevance . . . offer important opportunities for the evolution of values that may be necessary for transformative change towards sustainability" [15] (p. 1).

*Flotsam* [16] is a wordless book by David Wiesner. The plot of this illustrated book follows a camera that is deliberately tossed into the sea. Characters who find this camera take pictures with it, and then return it to the sea. For example, the main character is shown in the early pages throwing the camera he discovered back into the ocean. The layout of the images establishes a narrative that is easily interpreted without words. The first of three small panels on top of the spread shows photographs laying spread out on the beach, the second illustrates the character pensively looking at the camera, and in the third panel, the character swings his arm back with the camera strap in hand. "Flotsam" is also defined as the material in our oceans that was not directly tossed away, but through boat crashes or accidents has ended up in the sea.

The after-school programme is located within a short drive of the ocean, and all the children would be familiar with it; however, the sandy beaches portrayed in the book would not be a shared experience of all the children because beaches in near North Newfoundland and Labrador are naturally rocky. Even though the setting might be unfamiliar to all, through their discussion about this book, their experiences from home such as family beach holidays and school studies around oceans, the children discussed the animals they saw, the fantastic realities that were presented, and the way this all ties into the preservation and conservation of our oceans. We use pseudonyms for the children, pre-service teachers and early childhood educators. Take this excerpt from a focus group conversation between children and teachers at the centre as an example that illustrates how children connect their prior knowledge to an emotional understanding through a literacy inspired activity in a makerspace:

**Delores the after-school ECE:** "He's throwing it (the camera) away for other people to find"

**Jessie:** "You shouldn't be polluting, mister!" (Looking directly at the character in the book.)

**Mya:** "These people used their creative license completely"

**George:** "There's lot of pollution in the ocean" (Pointing at the pictures.) [39]

Children were asked to use the makerspace to create maker projects that responded to the clean-up of the ocean floor, or to consider the conservation and protection of the arctic ocean based on their socio-cultural learning experiences. Projects and conversations about individual pages of the story showed that children used their prior knowledge as a tool, often referencing other third spaces where

learning took place, such as school field trips to the local science museum, summer vacations with family, and other books and digital resources. Delores shared "I did not know that Mya's grandmother lives in Thailand and that she had been there as a small child" [40]. These experiences were entangled as part of the learning project alongside their crayons, paper and 'makey-makey' parts, which are technology kits the children used, such as electronics like circuits or LEDs, which are relatable to basic engineering components.

In this illustration shared with the children, the main character sits on the sand looking at a camera covered in algae which has just washed up on the shore. He wears brightly colored clothing, and the bucket and shovel beside him are vibrant in comparison to the muted blues and greys of the camera that has captured the underwater ocean. Children discussed the illustrations like this one with vigor. By allowing children the freedom to discuss what they see, they are also strengthening the ways that they connect to literature, their instructors, their external environments, their personal opinions and thoughts, and their peers. Through conversations like the one shared below, children begin to understand that conservation occurs in a community—that while individuals might strive for change, change does not happen in a vacuum.

**Mya:** "It's a picturebook, you can make up stories over and over"

**George:** "There's stuff in the water that's not supposed to be there"

**Delores:** "Can you see what's on the camera there?"

**George:** "Is it algae?"

**Corina:** "What kind of facial expression does he have?"

**George:** "Surprised! Because its polluted."

**Jessie:** "Moving (with voice excited and louder) . . . MOVING" (reference to the underwater garbage and human like sea creatures they are analyzing in each panel)

**Mya:** "Maybe they are trying to develop a human lifestyle"

**George:** "No they're going to grow up acting like a person in water"

**Jessie:** "They are acting too much like a human now!" (referencing the facial expressions of the humanistic algae) [39]

The children were encouraged to draw from prior knowledge to engage in critical discussion and formulate opinions around ocean pollution as a foundation for the makerspace—in which being a third space offered the researchers the opportunity to frame data within Oldenburg's framework. Kumpulainen explains that makerspaces "prescribe a model of learning-by-doing in which individuals can work on creative design projects that are personally and/or collectively meaningful. The possibility to play with material objects is considered to act as 'a social glue' for people to come together and engage in collaborative and creative endeavors" [20] (p. 14). By this definition, making activities unite people of all ages, genders, social levels, education, and expertise and build complex sets of creative and social systems within a third space. Children extensively interacted with the picturebook, and extended their knowledge to a field trip visit to a local science centre exhibit. Much of the flotsam they viewed was projected in their healthy as well as polluted habitats that mirrored the personified sea creatures seen in the book (see Figure 1).

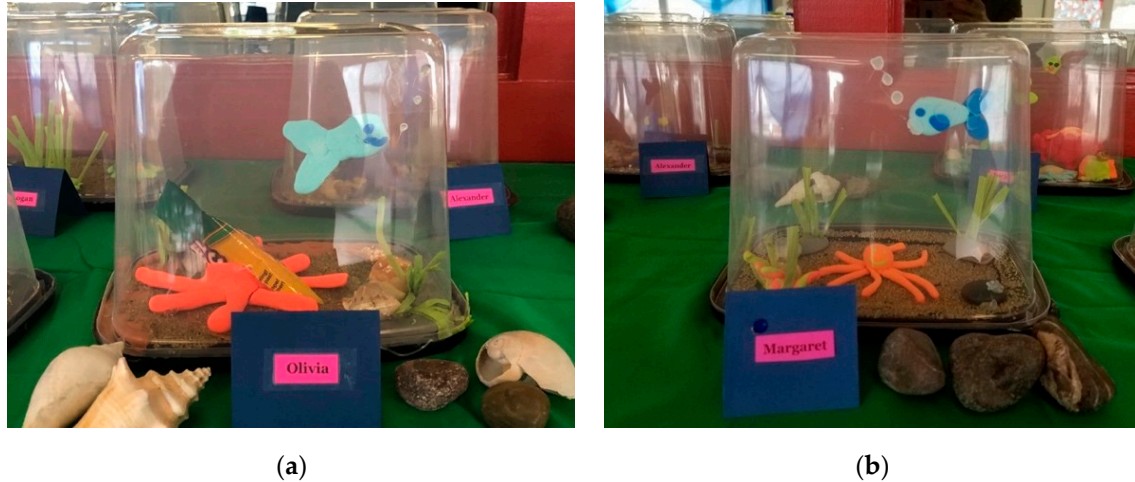

**Figure 1.** (**a**) An example of a "polluted" environment; (**b**) an example of a "healthy" environment.

After this activity was finished, in an effort to provoke more play and discussion about the ocean ecosystem, the teachers sprinkled garbage in the make-believe ocean created from playmaking, and on their biodomes. They asked children how they felt about it, and what the possible consequences could be as a result of human-created pollution. This spawned an excited dialogue about the ways they could lower or eliminate the pollution from the ocean. George shared, "We could make a book with pictures of animals saying 'don't pollute us'". Mya suggested that they use real photographs for their "Don't Pollute Us" book [39]. We saw how it created an empathetic connection between children and the undersea environment they were creating, furthering critical thinking about how human action impacts our ocean floor which is often invisible. Children's deep engagement within the makerspace project shows a building of knowledges around relational values concerning the ocean and their choices. Engagement of this type shows an active understanding of citizenship and environmental awareness within the makerspace as a third space when viewed through a sociocultural lens as defined by Oldenburg (see Table 1).

A key component to Oldenburg's Third Place theory is that people within the space accomplish an unwritten objective. The table above presents the characteristics that identify a third place, and when coalesced together, they help develop relationships between participating individuals within them. "Third places are face-to-face phenomena. When you go to a third place you essentially open yourself up to whoever is there" [41] (p.1). The after-school centre's mix of ages and demographics is unique for many of the children, who otherwise move between the classroom and home, both of which limit their interactions. The makerspace's materiality and informality allow children to create relationships. This third place is different as it is not just a makerspace, but also a learning ground for training teachers to understand how to make meaning and understanding the environment through play and digital literacies. It is also a place where training teachers can witness the development of student relationships between each other and environmental sustainability.

We draw from Chan, Gould, and Pascual (2018), who "propose a widening view of values to extend beyond the worth of nature itself (intrinsic values) and what nature does for us (instrumental values), to include preferences, principles and virtues about human-nature relationships (relational values)" [15] (p. A1) that are enmeshed within a third place. In this study, the use of children's environmental literature gave way to conversations among the children about sustainability and empathy for sea creatures. When learning about sustainability, Cheng and Monroe (2012) suggest that "children's perception of connection to nature consists of enjoyment of nature, empathy for creatures, sense of oneness, and sense of responsibility" [42] (p. 43). Importantly, their research coalesced in an index that outlined five factors that can predict interest in children's environmentally friendly practices: "family values towards nature, previous experiences with nature, nature near home, knowledge of environment, and a perceived self-efficacy" [42] (p. 44). When combined, these factors contribute to a

child's connection to nature, which they interpret as "attitude" [42] (p. 44) and which can be evidence of Chan, Gould, and Pascual's theories surrounding relational values that contribute to environmentally friendly practices. We found many of these sociocultural practices to underlie much of the data evidenced in how the children and the pre-service teachers were interested in ocean sustainability (see [43]).

**Table 1.** Descriptions of Oldenburg's characteristics of third places [14].

| Characteristics | Third Places Description |
|---|---|
| Neutral | "In which none are required to play host, and in which all feel at home and comfortable" [14] (p. 22) |
| Leveling | "By its nature, an inclusive place and does not set formal criteria of membership and exclusion" [14] (p. 24). |
| Conversational | "Nothing more clearly indicates a third place than that the talk there is good; that it is lively, scintillating, colorful, and engaging" [14] (p. 26) |
| Accessible and Accommodating | "Render the best and fullest service are those to which one may go alone at almost any time of the day or evening with assurance that acquaintances will be there … Correspondingly, the activity that goes on in third places is largely unplanned, unscheduled, unorganized, and unstructured" [14] (pp. 32–33). |
| Regular Attendees | "Give the place its character and who assure that on any given visit some of the gang will be there … Every regular was once a newcomer, and the acceptance of newcomers is essential to the sustained vitality of the third place" [14] (pp. 33–34). |
| Low Profile | "Third places are unimpressive looking for the most part. They are not, with few exceptions, advertised; they are not elegant" [14] (p. 36). |
| Playful | "The persistent mood of the third place is a playful one … Here joy and acceptance reign over anxiety and alienation." [14] (p. 37) |
| Home-Away-From-Home | "A congenial environment" [14] (p. 39) that "roots us" [14] (p. 39), "is appropria[ted]" [14] (p. 40), "regenerates and restores" [14] (p. 41), provides feelings of "being at ease or the 'freedom to be'" [14] (p. 41), and finally "warmth" [14] (p. 41). |

*4.1. Developing Relationships with Nature through Play*

Oldenburg argues that neutrality contributes to unity [14] (p. 23). This was also evident in the reactions of children when creating their biodomes. Peppler et al. argue that making is as much a process of Do-It-Yourself as it is Do-It-With-Others because projects are open-ended with both high-tech and traditional tools in confined spaces at participant disposal, 'makers' are introduced to multiple ability levels, differing project types, and 'maker' techniques [44] (p. 27). For makerspaces to be successful, all individuals must feel like comfortable participants in the space and process without being confined or limited in space or goal—much like the young children gathered at the after-school programme. Neutrality encouraged participation for children of different ages, genders, races, and economic classes to participate towards the creation of a single goal—how to preserve ocean life. The result of this was that children built an emotional connection to nature through this project that Cheng and Monroe (2018) argue is the "strongest independent variable that influenced children's interest in participating in nature-based activities" [42] (p. 44). The wordless award-winning book *Flotsam* [16] was also a neutral book in that it had no words, and required no explanation beyond the illustration, and was open to wide interpretations. Children with differentiated learning abilities felt welcomed to participate in the discussion because they could build upon what they already knew—offering prior learning experiences in conversation regarding pollution in the ocean, and the way they interpreted the images on the page. This was further observed with the way in which children enacted imaginative play through becoming sea horses (as seen in the book) entangled with materials

to mimic the danger of pollution on the ocean floor. These playful activities invited other children playing nearby to join in as puffer fish floating as royal and regal creatures.

To emphasize this point, two children who engaged in this play episode decided to create biodomes that showed both polluted and healthy habitats and worked together to draw connections and stimulate deep learning (see Figure 1). *Flotsam* [16] and the discussions about it, encouraged children to build emotional connections not only with nature, but with other children in the programme—connections that Chan, Gould, and Pascual (2018) argue are key components of relational values [15], and that Cheng and Monroe (2012) suggest also strongly contribute to "an opportunity to enhance children's affective attitudes toward nature and their interest in protecting nature" [42] (p. 45) that would not have been possible without first enacting this learning within a makerspace.

### 4.2. Conversation and Accessibility Creates Relational Values

While the goal to create a habitat was established from the outset, children were encouraged to create their own biodomes that illustrated their interpretations of the ocean using tools and objects available to them. The process of play-based learning for children in the makerspace was unstructured and wholly their own. Children had to manage their own time and derive their creations from their own inspiration based on the discussions while reading the book *Flotsam* [16]. Children showed their understanding of environmental sustainability to others through the connections they drew from conversations about *Flotsam* [16]. Letting go of control in a third space, according to Oldenburg's component of *accessibility and accommodation* is necessary for the success of the individuals who participated within it [14]. We saw this reflected through the actions and creations of the children's habitats. We were fascinated with the knowledgeable opinions that children shared and prior experiences with the ocean—previous experiences that Cheng and Monroe (2018) argue are critical towards developing long-lasting conscientiousness with nature [42]. This led to personal investment in the creation of their biodomes, which in turn—when pre-service teachers 'polluted' them with garbage (pieces of paper, corners of plastic bags etc.)—developed empathy and an emotional relationship with ocean conservation that connected the relational and moral principles showing that excessive and careless pollution of the oceans is wrong, but that there can be ways humans and nature can coexist harmoniously. Teachers shared that deep discussion led to complex questions, which contributed to children connecting their learning about conservation and sustainability to other topics. Two pre-service teachers shared:

"We allowed children to develop their own inquiries and build communal understanding based on their perceptions. Once children created a communal understanding of ocean habitats, they collaborated in small groups to put this information into action. By showing and sharing their projects children were able to apply their gained knowledge and refine their questions. This refinement resulted in more complex questions which led to a better understanding of pollution and ocean habitats" [45].

Chan, Gould, and Pascual (2018) state that "values that are *relational in content*, that is values where the relationship *itself* matters, as more than a means to an end (a preference for seeing birds is relational in origin, a sense of kinship with birds is relational in content)" [15] (p. A4). Relational values like these can partially be observed through conversation expressed by children in this project, which was a key element to the success of children's understanding of sustainability aided by the context of makerspaces and the literacy learning that happened within them.

### 4.3. When Children Cannot Learn Directly from the Sea

St. John's, Newfoundland, is a seaside community. Resting comfortably on the shores of the North Atlantic Ocean, the city boasts a deep, sheltered harbor at the base of several large hills. Because of the limitations of transportation being offered at the after-school centre, *regular attendees* are unable to visit the ocean directly during our project, but they could interact with concepts such as conservation through literacy learning, digital media, and creativity, which was a part of the curriculum unit. Inclusivity in this way is imperative, because children can build on and learn from others with

different experiences and opinions resulting in a shared understanding of the concept of environmental sustainability. New children and new ideas were welcomed through this project, because it inspired other children to speak up, voice their ideas, and make new connections which sparked curiosity. Children who knew each other often voluntarily worked together on a number of ocean-inspired projects throughout the 6 weeks. This comfort was evident in their creation of conflicting habitats represented in their biodomes demonstrating how the open-ended makerspace acted in the capacity as a third space, bringing together vast knowledges and experiences of the children.

The maker literacies that evolved within this makerspace was informal and provided the opportunity for children use their own voices and opinions, which enhances the learning experience, as expressed through relational values that Chan, Gould, and Pascual (2018) described [15]. In fact, Oldenburg's characteristic of *low profile* contributed positively to the comfort levels of the children who offered opinions and suggestions about cleaning up the oceans. If the experience were somewhere more formal, like a lab or museum, perhaps the children would not have felt as comfortable talking as much as they did. For example, the physical makerspace utilized for the project on oceans for these children varied compared to other groups who embraced digital apps. The makerspace was just a spare room with tubs and loose parts on the table—clay, sequins, rocks, tissue paper, and glue. But children built deep learning connections between what they knew and what they wanted to know and bridged gaps such as play making different sea creatures, and corals during free play. We saw results much like Barthel, Belton, Raymond, and Giusti's describe as a positive change and "increase concern, interest in and/or care for nature" [21] (p. 6). We saw this concern expressed from children in this makerspace, especially when pre-service teachers "littered" their make-believe ocean and later their biodome projects with shreds of paper and old plastic wrappers. Children responded viscerally to this action, but their shock was replaced by realization, understanding, and empathy. They equated their feelings of dismay when they were "polluted" with the feeling of the fish. Children expressed the relational values of context conflated with the relational values of content to conclude that they would not pollute the ocean because they knew what pollution *felt like.* Making literacy moments such as these would not have been possible without the third space providing a low-pressure environment where children could build emotional attachments to their work without worrying their learning would be seen as "wrong" by their peers or teachers.

## 5. Areas for Future Research

This makerspace project connected children to learning in a congenial environment—one that encouraged conversation in positive ways, and that was both warm and thoughtful. Children felt that their experiences were valued by their peers, their early childhood educators and the pre-service teachers in training. Moreover, because of the convivial atmosphere, children found worth in personal terms in their learning and in the biodome projects that showcased their learning. Conversations with both parents and children about the project revealed that because the learning happened in an environment with peers and teachers, and was aided with many supports that felt like home, children gained new knowledge to share with family and friends, thus expressing relational values between their learning contexts and home contexts.

Lastly, the connection that other scholars have made between home and school, using literacy as a third space, is one that cannot be understated. Cook (2005), Pahl and Kelly (2005), and Brooker (2011) assert that literacy creates a third space where children can make connections by drawing from their 'Funds of Knowledge' [46] between home and school [47–49]. In this case, applying Oldenburg's analysis to our data in connection to Chan, Gould, and Pascual's (2018) discussion of relational values, our research shows that the makerspace can be the third space that provides the opportunity for children to make connections between in-school and out-of-school learning that develops empathy that can "change children's affective attitudes toward nature and their interests in protective nature" [15] (p. 45).

While this research found that third spaces created through makerspace opportunities for young children were fruitful, it was limited in its scope as it focused on only one early childcare centre.

One avenue for future research would be the application of Oldenburg's theories within multiple early childhood centres, observing how the differences in each setting manifest, and considering how the sociocultural characteristics of Oldenburg's third space would provide greater insight into the importance of interrelationships of communities, space and place.

Furthermore, consideration of the ways that makerspaces in public areas such as mobile labs, parks, museums, libraries, or science centres also work as third spaces would be a valuable area for future research, following Potter and McDougall's observation that such study requires a framework [23]. Oldenburg's work from 1997 provides such a framework for the use of third spaces within early childcare learning centres—though it has not been applied here before this article. It is important to consider this framework, and how makerspaces as a third space encourage deep learning for young children in spaces that are already labeled as third spaces by Oldenburg, such as libraries, museums, and science centres.

## 6. Conclusions

To conclude, young children, when confined to a classroom, develop connections with nature by experiencing empathetic growth. Though the children in the early learning centre could not physically visit the closely situated Atlantic Ocean during this project, they began to understand the importance of connecting their personal daily activities to the ways they can impact the world around them. Makerspaces, specifically when viewed as third spaces that can foster low-risk freedom to explore ideas, can greatly impact the way children connect with the natural world around them. We saw this materialize when two children were able to compare and contrast a "healthy" undersea environment with a "polluted one". These biodome projects were created by the children and reflect not only a physical connection to the world, but also develop positive emotional and empathetic connections that deepen the desire within oneself to protect and conserve something loved.

The Early Childhood Educators and pre-service teachers also played an integral role in this makerspace project. They facilitated imaginative learning possibilities as they nudged children to build connections based on the books they read and the biodomes that they created. The pre-service teachers' playful teaching around pollution and trash forced students to consider the perspective of animals in the ocean, and that perhaps those animals might feel similarly towards having their environments polluted. Though not considered an educational researcher, Oldenburg's work on third places provides a framework for how these pre-service teachers and children learned in a makerspace. This project's emphasis on maker literacies in makerspaces introduced children to a learning experience that was not radically different from the types of exploration seen outside of school or in the home, or in a third space. Importantly, the project's third space is one that provided children a way to explore their world around them in meaningful and empathetic ways.

This research answers Potter and McDougall's observation about school contexts that "What was lacking . . . was a way of bringing the material and the virtual together in a commentary which attempted to fully account for digital making in a third space from a sociomaterial and sociocultural perspective" [23] (p. 53). Without applying Oldenburg's theory, it would not be evident why this makerspace contributed so greatly to the children's learning process. This project shows how the type of literacy learning children enacted within a third space encouraged them to connect past learning with their present, in order to create curiosity for future learning opportunities.

Furthermore, we conclude that the context in which children learn is one of the most important factors that facilitate deep learning experiences. Third spaces that exist in between home and school provided children the freedom to connect ocean sustainability and conservation through maker literacies that lead to a deeper emotional understanding. This unique undertaking of encouraging children's voices as instrumental in our research shows that when children are given explorational freedom to develop their own narrative learning journeys, they found relational values that positioned them as conservationists. Not only were young children able to articulate their prior understanding of the natural world through play and maker literacies but they also built empathetic relationships with

the ocean and the organisms living in it. These two components of our research—the physical and the emotional—are not mutually inclusive, but when connected affirmed the success of a third space's ability to foster within it the deepest and most valuable learning experiences for its participants.

Playmaking and multimodal exploration with the materials in the makerspace led to strong voices of responsibility on the part of the children to upkeep our oceans. Those feelings of pride, responsibility, and ownership paved the way for children to feel both empathy and empowerment. These transitional feelings blossomed when children realized organisms in the sea must feel similarly, when the habitats they love and live in are polluted. When children embodied creatures in the ocean during their playmaking time, we saw how the thread of imagination that ties the children together and their experiences with ocean sustainability created a bond that established the children as conservationists and contributed to their environmental identity, thus building relational values.

**Author Contributions:** Conceptualization, A.B.; Data curation, A.B. and A.C.; Formal analysis, A.B. and A.C.; Funding acquisition, A.B.; Investigation, A.B. and A.C.; Methodology, A.B.; Project administration, A.B.; Supervision, A.B.; Writing—original draft, A.B. and A.C.; Writing—review & editing, A.B. and A.C. All authors have read and agreed to the published version of the manuscript.

**Funding:** This work was supported by the Social Sciences and Humanities Research Council of Canada Connections Grant (611-2018-0449) and the Memorial University Office of Public Engagement.

**Conflicts of Interest:** The authors declare no conflict of interest.

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
