# Peer review of "“Making” Waves: How Young Learners Connect to Their Natural World through Third Space"

_education, doi:10.3390/educsci10080203_

Round 1

Reviewer 1 Report

Your study got my fullest attention; I was intrigued by the well summarized and sound abstract. The topic is of high relevance and provides an iIt was

Your study got my fullest attention; I was intrigued by the well summarized and sound abstract. The topic is of high relevance and provides an interesting perspective by exploring the linkages between the makerspace construct with multimodalities and third place theory. 

However, the study needs major methodological revisions; as a reader, I kept waiting to know what happened at the early childhood centre and how participants (children, teachers, and ECes) engaged with the study. It is not clear, for example, if the participants were ECEs or the children, further it was confusing to understand if the study happened at an after school program at an early childhood centre or if the place was a combination of both. I would recommend defining these institutions within the jurisdiction of where the study took place. 

It was also confusing to understand if the children live by the ocean (lines 115-117) or not (lines 495-496). I believe this data is fundamental to understand the study and the participant's situatedness.

What question(s) guided the study? How were participants recruited? How and by who was data collected? For how long? 

The case study approach also needs major revisions; I would recommend reading Creswell (2015), Yin (2015), Merriam (1998) and Stake (1979, as cited in Denzin & Lincoln, 2000, p. 437)

The literature is too long, and the central argument gets lost; I would recommend summarizing critical aspects of it. 

Little evidence is provided; there is not enough data to draw the findings and conclusions. 

The contributions of this work could help scientists to advance and to find the best ways to connect education with ecology. I think the paper has the potential to attract readers if significant revisions are considered. 

Author Response

Your study got my fullest attention; I was intrigued by the well summarized and sound abstract. The topic is of high relevance and provides an iIt was

Your study got my fullest attention; I was intrigued by the well summarized and sound abstract. The topic is of high relevance and provides an interesting perspective by exploring the linkages between the makerspace construct with multimodalities and third place theory. 

However, the study needs major methodological revisions; as a reader, I kept waiting to know what happened at the early childhood centre and how participants (children, teachers, and ECEs) engaged with the study. It is not clear, for example, if the participants were ECEs or the children, further it was confusing to understand if the study happened at an after school program at an early childhood centre or if the place was a combination of both. I would recommend defining these institutions within the jurisdiction of where the study took place. 

  • In consideration of this feedback, we more thoroughly defined the institutions involved on pages 5 and 6 under Section 3: Our Research. We provided descriptions of who participated with us in our research. Describing what the Early Childhood Centre looks like and the experiential learning that happens within it, is a critical edit to this piece. We have not only brought together the full extent of the participants involved and the setting in which learning took place, but the theoretical reasons behind the importance of connecting explorational learning has to the space it is enacted.

It was also confusing to understand if the children live by the ocean (lines 115-117) or not (lines 495-496). I believe this data is fundamental to understand the study and the participant's situatedness.

  • To clarify this point, we have provided a more detailed description of the local geographic context in which the early childhood centre is located – St. John’s, Newfoundland on page 8 and revisited again in section 4.3. We’ve reframed the paper to discuss how the context and setting of the makerspace at the early childhood centre provide the appropriate situatedness for children to take control of their own learning and develop empathy in both individual and communal forms. Our new focus allows for a connection to be drawn between the local setting of Newfoundland, the Early Childhood Centre, and how children can use literacy to draw connections between their own personal experiences and the images they see in a storybook through “making”.

What question(s) guided the study? How were participants recruited? How and by who was data collected? For how long? 

  • In section 3.1 we have included an entire discourse about our Research Ethics. Our ethics were based on Lundy’s Child-Centred Learning Model drawn out of the United Nations Convention on the Rights of the Child, and the Right of the Child to be Heard. Secondly, in Section 3: Our Research, we emphasize that the research project occurred between 2018-2019 for six weeks. We spent 3-6 hours a week at the site resulting in 18-36 hours total research tine. We’ve included in Section 3 the relationship that participants have to the Early Childhood Centre, and in Section 4.2, we’ve included how Pre-Service Teachers felt about their own experiences in participating with the project by referencing their reflection journals.

The case study approach also needs major revisions; I would recommend reading Creswell (2015), Yin (2015), Merriam (1998) and Stake (1979, as cited in Denzin & Lincoln, 2000, p. 437)

  • We have included a discussion in section 3.2 about how we consider our Case Study as qualitative rather than quantitative research. We consider Yin (2009), Given (2000), Creswell (2013), Kridel (2010), Denzin & Lincoln (2000), and Hertz (1996). Their discussions about including children’s voices within a case study have supported the theoretical frameworks we reference in section 2. This allows us to draw connections between how impactful context can be to help children develop identities that connect empathetically to the natural world around them.

The literature is too long, and the central argument gets lost; I would recommend summarizing critical aspects of it. 

  • We have reframed the theoretical narrative by cutting much of the critical aspects of our literature review. We have included specific discusssions of how makerspaces are structured (Section 2), the learning that is enacted within makerspaces, (Section 2.1 and 2.3), how we are framing a Third Space along with a brief description of what a third space is and how it is connected to our research (Section 2.2). This discussion encompasses only 4 of our existant 18 pages and it is key to give a firm theoretical foundation so that readers understand why we are framing our research in this lens.

Little evidence is provided; there is not enough data to draw the findings and conclusions. 

  • With our new focus through Lundy’s Child Centred model, section 4.3 is a critical addition to our paper that highlights how pre-service teachers viewed their participation and their reactions to the participation of the children. We have provided responses from a reflection journal to emphasize our points further. We have also included more nuanced discussion of children’s biodomes in section 4.3 as evidence for children developing empathetic connections to nature through learning in a third place makerspace. In section 4, we shortened our descriptions of Flotsam and replaced it with connecting analysis of how children’s conversations evidenced their not only how children expressed their learning, but their ability to take control over their learning by providing new insight on the part of the Early Childhood Educator.

The contributions of this work could help scientists to advance and to find the best ways to connect education with ecology. I think the paper has the potential to attract readers if significant revisions are considered. 

Reviewer 2 Report

My concerns regarding the present study revolves around several parts. Firstly, I was not able to detect any explicit research question(s). The three sentences, in the lines 44,45,46, by which are somewhat repetetitive, inform about the research and the content of the article, but I miss information about what specific research questions guided the study. The research questions and aims should be clearly presented, as the reader is not supposed to spend a lot of time wondering what the study specifically aims at.

The authors conclude that, Oldenburg's third place theory can be well applied to research within the education sphere. Further, that the research answers Potter's and McDougall's observation etc., and finally, they conclude that the context in which children learn is one of the most important factors that facilitate deep learning experiences. Not being familiar with neither Oldenburg's third place theory nor Potter's and McDougall's observation, they are presented and outlined in the text and thus I get an impression that the conclusions are thrustworthy, however, the lack of specific research question is disturbing.

Secondly, but most important, a chapter concerning research ethics is completely missing. The authors mention ethics once, in line 260, where they claim that all participants gave their informed consent. How did the children give their informed consent? How did the researchers go about quality assuring that the children understood what they agreed to? Issues of conducting research with young children are very well outlined and discussed in several research articles, and whether children are able to give an informed consent is debated. Only gaining consent from their parents, and thus conclude that everything is fine, is not sufficient when conducting research with children. Researchers that involve young children have a particular responsibility to safeguard their participants, this includes their dissent to participate, regardless of whether their parents have given their consent. What kind of ethical considerations did the researchers discuss? 

Finally, the structure and logic of the text must be improved, e.g. chapter 2, has two sections that refer to 2.1; Maker literacies, and Manners of learning. Chapter 3 present the case study. What is a case study and how does this research refer to being a case study? This is not clearly presented or argued. What is the case under study? - how many participants? Perhaps is it not necessary with major revisions concerning this in particular, but clarifications are definetely needed.

Author Response

My concerns regarding the present study revolves around several parts. Firstly, I was not able to detect any explicit research question(s). The three sentences, in the lines 44,45,46, by which are somewhat repetetitive, inform about the research and the content of the article, but I miss information about what specific research questions guided the study. The research questions and aims should be clearly presented, as the reader is not supposed to spend a lot of time wondering what the study specifically aims at.

  • We have included our two research questions at the end of section 1: How could maker literacies, materials and play-making in a makerspace help children build a relationship with nature, and how can a makerspace as a third space aid children’s emotional connections and responsibility for ocean conservation?

The authors conclude that, Oldenburg's third place theory can be well applied to research within the education sphere. Further, that the research answers Potter's and McDougall's observation etc., and finally, they conclude that the context in which children learn is one of the most important factors that facilitate deep learning experiences. Not being familiar with neither Oldenburg's third place theory nor Potter's and McDougall's observation, they are presented and outlined in the text and thus I get an impression that the conclusions are trustworthy, however, the lack of specific research question is disturbing.

  • We re-adjusted our literature review so that Oldenburg’s theories are more prominently discussed in relation to how his theories connect to maker literacies (Sections 2.1 and 2.2). We included our research questions at the end of Section 1: Introduction so that readers have a clear understanding of what our paper discusses, and how it is framed within the confines of our research. Further, in our discussion of Ethics (3.1) we discuss Lundy’s model of Child Centred Research and include this as an important discussion that paves the way for our discussion of children learning in collaboration in a makerspace within the freedom of a third space.

Secondly, but most important, a chapter concerning research ethics is completely missing. The authors mention ethics once, in line 260, where they claim that all participants gave their informed consent. How did the children give their informed consent? How did the researchers go about quality assuring that the children understood what they agreed to? Issues of conducting research with young children are very well outlined and discussed in several research articles, and whether children are able to give an informed consent is debated. Only gaining consent from their parents, and thus conclude that everything is fine, is not sufficient when conducting research with children. Researchers that involve young children have a particular responsibility to safeguard their participants, this includes their dissent to participate, regardless of whether their parents have given their consent. What kind of ethical considerations did the researchers discuss? 

  • We’ve included Lundy’s child-centric learning model (3.1) in the description of our research ethics. Further consideration towards the right for a child to direct research was discussed in connection with makerspaces and third places as settings that can best facilitate a child’s right to direct their learning. We connect Lundy’s model of ethics that situate the child as a leader in their own research as an imperative component to learning within a makerspace and within a third space. We’ve also included examples from pre-service teacher participants in our research that express their observations about how children learned within the makerspace as further evidence supporting our conclusions that through third places and makerspaces, children are empowered to draw from their previous experiences, narratives, and opinions, find value in them, and then use these validated experiences as firm foundations to foster senses of deep empathy towards concepts like ocean sustainability.

Finally, the structure and logic of the text must be improved, e.g. chapter 2, has two sections that refer to 2.1; Maker literacies, and Manners of learning. Chapter 3 present the case study. What is a case study and how does this research refer to being a case study? This is not clearly presented or argued. What is the case under study? - how many participants? Perhaps is it not necessary with major revisions concerning this in particular, but clarifications are definetely needed.

  • We shortened Section 2 and split it into our three core concepts: maker literacies (2.1), and third spaces (2.2), and manners of learning (2.3). Structuring our work in this way allows us to build a bridge between maker literacies and third spaces with the ways that children learn in each space – ultimately intimating that makerspaces are third places where learners can easily take control of their own learning journeys.
  • We added in a section that discusses our qualitative case research (section 3.2). We consider Yin (2009), Given (2000), Creswell (2013), Kridel (2010), Denzin & Lincoln (2000), and Hertz (1996). Their discussions about including children’s voices within a case study have supported our theoretical framework that allows for us to explore how context impacts children’s learning and empathetic identity development.
  • We have also explained the timing of the study in Section 2 and on page 6. A more detailed description of what the setting of the project entails and who was involved on page 5.

Reviewer 3 Report

Review of the manuscript

 ”Making” Waves: How Young Learners Conect to their Natural World through Third Spaces

I have with great interest read this manuscript and have a few comments of it. It is an interesting topic, with possibility to give further knowledge of issues like use of maker spaces in teaching, possibilities of third space and how these tools in education give children further possibilities if learning. This manuscript is though at a level where this is not happening yet, but with a lot of work put into processing the paper that can be possible. The structure of the paper is very poor at the moment, and this makes the manuscript very hard to grasp and the main points hard to unravel. I am highlighting central issues, not details, since the structure of the paper needs to be improved first.

I will give a few examples of what need to be done to improve the readability of the manuscript.

The introduction is introducing many concepts and standpoints that are going to be developed further on. The text is at the moment too compact and lacks discussion of the concepts introduced, or they are discussed in a way not clearly linked to other concepts introduced. An introduction should introduce the theme(s) of the manuscript clearly making it clear to the reader what comes next. This is not really happening here. In the second paragraph of the introduction there are some valuable arguments that could be built on but then the author also should stick to these thorugout the manuscript, which is not the case at the moment.

In the introduction the concept ‘Third Space’ is introduced without reference, and such introduction combined with definition comes several times in the manuscript. These should be kept together. This is a problem thorughout the text, that the lack of references when discussing crucial concepts or referencing to studies makes the liability of the reasoning low.

The part of the manuscript where makerspaces is discussed is interesting but har to follow. This should be structured with a clearer mind from the author what story he/she  wants to tell. Further in this section in paragraph 2.1 there is a short description of the study conducted, and a description of Third Space. These sentences should be placed in sections describing the study conducted and a definition of Third Space. The definition of Third Space should also be theoretically related to Makerspaces.

In the section describing the case study there are theoretical reflections that are hard to understand why these are placed here. These should be put together with other theoretical definitions. And descriptions of the case study should be placed here. The description of the case study need to be more precise and accurate, and aspects like research ethics and liability need to be discussed.

The empirical examples are not clear why these are displayed and how these strengthen the understanding of the case. This might be linked to the fact that there is no clear research questions presented in the manuscript, and the aim is weak.

The section 4 need to be presented in written text and these should be presented more in detail. Oldenburg’s concepts are hard to understand in this context, unfortunately.

The presentation of affordance theory in the empirical section is very hard to understand why it is presented there and not anchored to the other theoretical concepts that seems to be focused on in this manuscript.

In order to make the discussion clearer and more linked to the earlier sections in this paper, all these other issues need to be dealt with before restructuring the discussion, and making it more clear what issues the author/s want to raise.

Author Response

The introduction is introducing many concepts and standpoints that are going to be developed further on. The text is at the moment too compact and lacks discussion of the concepts introduced, or they are discussed in a way not clearly linked to other concepts introduced. An introduction should introduce the theme(s) of the manuscript clearly making it clear to the reader what comes next. This is not really happening here. In the second paragraph of the introduction there are some valuable arguments that could be built on but then the author also should stick to these thorugout the manuscript, which is not the case at the moment.

  • We tied the introduction (Section 1) to the literature review more succinctly by providing our two research questions. We then supported these questions with a theoretical discussion (section 2) to provide background for our research analysis. Our literature review is more concise, including only three theoretical concepts which build upon each other and provide a firm foundation for the reader to consider the implications our of our research analysis (Section 2.1, 2.2, 2.3), We consider how physical connection to a task develops emotional connection and contributes to the creation of identity that makes children draw deep connections between creatures in the ocean and themselves. This connection is enhanced and discussed throughout the paper using Lundy’s Child-Centred Learning Model and the Right for a Child to be Heard – discussed in Section 3 and revisited throughout our analysis. By cutting down our theories, we’ve woven a thread throughout the sections of our paper to ultimately conclude that children’s abilities to take control of their learning in a makerspace that presents itself as a third space allows them to become more conscious conservators.

In the introduction the concept ‘Third Space’ is introduced without reference, and such introduction combined with definition comes several times in the manuscript. These should be kept together. This is a problem thorughout the text, that the lack of references when discussing crucial concepts or referencing to studies makes the liability of the reasoning low.

  • We introduce Oldenburg’s theories as section 2.2 in our paper. It is a crucial concept for readers to understand as it is referenced, and its impact on our research is repeated in our analysis (Section 4). To amplify our analysis, we have included a table of Oldenburg’s 7-part theory on page 10 as it is the appropriate place for readers to connect Oldenburg’s theories more closely with our analysis considering we draw directly from each of the seven terms mentioned. We have also made sure that when we make a specific reference to a concept or an author, that we cite the concept and author so that we maintain our research credibility (such as on page 2, page 9, and page 11).

The part of the manuscript where makerspaces is discussed is interesting but hard to follow. This should be structured with a clearer mind from the author what story he/she wants to tell. Further in this section in paragraph 2.1 there is a short description of the study conducted, and a description of Third Space. These sentences should be placed in sections describing the study conducted and a definition of Third Space. The definition of Third Space should also be theoretically related to Makerspaces.

  • In our review we considered what story we wanted to tell. To make sure that our story was streamlined, clear, and conceptually interesting, we condensed our theoretical discussion into three distinct parts that inform our research analysis. We separated our study and third space theories into two separate sections of this paper so that readers will not get confused by the points we are trying to make (Section 2.2 and Section 4 respectively). We’ve also reviewed our paper and considered other areas where we combined analysis with theoretical discussion. We’ve added a consideration of Lundy’s Child Centered Model to our research ethics (section 3.1) and shortened discussion of other studies by Potter and MacDougall (2017) so that our paper is more focused on an analysis of the research we conducted rather than on the research conducted by others. This practice has allowed us to draw connections between relational learning, the right of a child to be heard by others, and third space theory within a makerspace.

In the section describing the case study there are theoretical reflections that are hard to understand why these are placed here. These should be put together with other theoretical definitions. And descriptions of the case study should be placed here. The description of the case study need to be more precise and accurate, and aspects like research ethics and liability need to be discussed.

  • Upon review of this project, we included a discussion of our research ethics under section 3 “Our Research”. Our ethics (section 3.1) are focused around a children’s right to be heard following the United Nations Convention on the Rights of the Child and Lundy’s theories surrounding Child Centered and Child Led Research.
  • We have also included a discussion in section 3.2 about how we consider our Case Study as qualitative rather than quantitative research. We consider Yin (2009), Given (2000), Creswell (2013), Kridel (2010), Denzin & Lincoln (2000), and Hertz (1996). Their discussions about including children’s voices within a case study have supported the theoretical frameworks we reference in section 2. This allows us to draw connections between how impactful context can be to help children develop identities that connect empathetically to the natural world around them.

The empirical examples are not clear why these are displayed and how these strengthen the understanding of the case. This might be linked to the fact that there is no clear research questions presented in the manuscript, and the aim is weak.

  • We removed discussion of the illustrations of the book Flotsam in favor of adding analytical discussion about the children’s conversations that they had when reading it. We linked our analysis back to our research questions by continually referencing why and how children build relationships with nature through emotional connections in third places (page 8 and 9). We added more analysis about what Pre-Service teachers encountered by adding examples from their reflection journals that further support the children’s right to learn (Section 4.2) and we included images from children that build corresponding habitats based on their experiences with the preservice teachers in a third space (Section 4.3). Our empirical examples are strengthened by tethers directly to our theories that support the need for children’s rights within their learning environments bolstered by makerspaces. Together, we conclude, that these two teaching paradigms provide an excellent pathway for children to develop empathetic connections to the sea (Section 5: Conclusion).

The section 4 need to be presented in written text and these should be presented more in detail. Oldenburg’s concepts are hard to understand in this context, unfortunately.

  • We did not delete the table, as we believe that it provides the best way for readers to understand the seven characteristics of Oldenburg’s theory. However, we folded the table more completely into the analysis of our research by providing it as an example that supports our discussion of the theory and its relationship to our findings so that it isn’t its own section.

The presentation of affordance theory in the empirical section is very hard to understand why it is presented there and not anchored to the other theoretical concepts that seems to be focused on in this manuscript.

  • We moved Affordance theory to section 2.1 as part of the theories that support and discuss maker literacies. When we added in Lundy’s theories in our ethics section (3.1), these two tie our conclusion that children develop an ecological mindedness through “action in every setting” (Barthel, Belton, Raymond, and Giusti, 2018, 3). This small aside is key to understanding how third spaces and makerspaces can contribute when considered as a cohesive unit to children’s ability to develop their own identities and how they connect with nature (Section 5: Conclusion).

In order to make the discussion clearer and more linked to the earlier sections in this paper, all these other issues need to be dealt with before restructuring the discussion, and making it more clear what issues the author/s want to raise.

Round 2

Reviewer 2 Report

Well done! The article is significantly improved by the authors, and should be published.